# New Indazol-Pyrimidine-Based Derivatives as Selective Anticancer Agents: Design, Synthesis, and In Silico Studies

**DOI:** 10.3390/molecules28093664

**Published:** 2023-04-23

**Authors:** Hanaa M. Al-Tuwaijri, Ebtehal S. Al-Abdullah, Ahmed A. El-Rashedy, Siddique Akber Ansari, Aliyah Almomen, Hanan M. Alshibl, Mogedda E. Haiba, Hamad M. Alkahtani

**Affiliations:** 1Department of Pharmaceutical Chemistry, College of Pharmacy, King Saud University, P.O. Box 2457, Riyadh 11451, Saudi Arabiasansari@ksu.edu.sa (S.A.A.);; 2Department of Natural and Microbial Products National Research Center, El Buhouth Street, Dokki, Cairo 12622, Egypt; 3Department of Therapeutic Chemistry, Pharmaceutical and Drug Industries Research Division, National Research Center, El Buhouth Street, Dokki, Cairo 12622, Egypt

**Keywords:** indazol-pyrimidines, anticancer, cell cycle, apoptosis, computational studies

## Abstract

In this research study, the authors successfully synthesized potent new anticancer agents derived from indazol-pyrimidine. All the prepared compounds were tested for *in vitro* cell line inhibitory activity against three different cancerous cell lines. Results demonstrated that five of the novel compounds—**4f**, **4i**, **4a**, **4g**, and **4d**—possessed significant cytotoxic inhibitory activity against the MCF-7 cell line, with IC_50_ values of 1.629, 1.841, 2.958, 4.680, and 4.798 μM, respectively, compared to the reference drug with an IC_50_ value of 8.029 μM, thus demonstrating promising suppression power. Compounds **4i**, **4g**, **4e**, **4d**, and **4a** showed effective cytotoxic activity stronger than the standard against Caco2 cells. Moreover, compounds **4a** and **4i** exhibited potent antiproliferative activity against the A549 cell line that was stronger than the reference drug. The most active products, **4f** and **4i**, werr e further examined for their mechanism of action. It turns out that they were capable of activating caspase-3/7 and, therefore, inducing apoptosis. However, produced a higher safety profile than the reference drug, towards the normal cells (MCF10a). Furthermore, the dynamic nature, binding interaction, and protein–ligand stability were explored through a Molecular Dynamics (MD) simulation study. Various analysis parameters (RMSD, RMSF, RoG, and SASA) from the MD simulation trajectory have suggested the stability of the compounds during the 20 ns MD simulation study. In silico ADMET results revealed that the synthesized compounds had low toxicity, good solubility, and an absorption profile since they met Lipinski’s rule of five and Veber’s rule. The present research highlights the potential of derivatives with indazole scaffolds bearing pyrimidine as a lead compound for designing anticancer agents.

## 1. Introduction

Cancer is a polygenic disease and one of the major aggressive health problems facing humans [1]. A report published by GLOBOCAN 2020 estimated that 19.3 million cancer cases and almost 10.0 million cancer deaths occurred in 2020 [2]. High mortality rates were reported in lung and colorectal cancer globally, in addition to breast cancer [3]. Treatment of cancer causes a state of extreme physical or mental fatigue until now due to its strong toxicity and the lack of efficiency of commercial anticancer drugs. In recent years, effective targeting has led to the development of more efficient and less toxic anticancer agents [4]. Pyrimidine derivatives are one of the most effective bioactive agents that exhibit a wide range of medicinal applications because they are essential parts of the nucleic acids that make up DNA and RNA [5]. Pyrimidine derivatives are heterocyclic scaffolds that represent one of the most bioactive pharmacophores implemented as anticancer agents [6,7,8,9] by using different mechanisms. For example, pyrimidines act as selective dual inhibitors of c-Met and VEGFR-2 [10], dual ERα/VEGFR-2 ligands with anti-breast cancer activity [11], and selective inhibitors against triple-negative breast cancer cell line MDA-MB-468 [12], and they have antiproliferative activity and EGFR and ARO inhibitory activity [13]. Furthermore, some pyrimidines act by inhibiting different proteins and enzymes that play key roles in the cell cycle and division [14]. Also, they act as antituberculosis agents and antimicrobials [15,16]. On the other hand, indazole derivatives are one of the most outstanding scaffolds and have a wide range of properties, including anti-inflammatory, anti-HIV, antiplatelet, and serotonin 5-HT3 receptor antagonist characteristics [17,18,19]. Peer-reviewed literature has reported that indazoles possess antibacterial, anti-inflammatory, antitubercular, and antidepressant traits [20,21,22]. Indazole derivatives exhibit potential anticancer activity, which makes them useful scaffolds for the development of new anticancer agents [23]. Indazole-*N*-phenylpyrimidin-2-amine derivatives **I** were synthesized and demonstrated bioactive products by Pawel M. et al. (Figure 1) [24]. CYC116 (Figure 1, **II**) is a novel anticancer molecule targeting both the cell cycle and angiogenesis, with antitumor activity in both solid tumors and hematological cancers [25]. Elsayed et al. [26] synthesized new indazole-pyrimidines (Figure 1, compound **III**) with potent antigenic effects against VEGFR kinase. Moreover, a sulfonamide moiety incorporated into different heterocyclic ring systems has been reported as one of the most specific scaffolds to inhibit the growth of different types of human cancer cells [27,28]. Thus, both pyrimidine and indazole derivatives represent privileged scaffolds in medicinal chemistry. Our strategy is to incorporate these two heterocyclic moieties in one molecule through an amino and/or sulfonamide linkage to synthesize new indazol-pyrimidine derivatives and evaluate their antiproliferative activity using an MTT assay against MCF-7, A549, and Caco2 cell lines. The most promising candidates were examined for their mechanism of action, their effect on the cell cycle, and their apoptosis stimulation potential on cancer cells, followed by their effect on normal cells. We are hoping to construct new bioactive heterocyclic hybrids that may enhance or increase the biological activities of the new products and may be useful scaffolds for developing new effective anticancer agents with high safety profiles. Finally, in silico studies, including molecular docking and drug-likeness studies, were performed.

## 2. Results and Discussion

### 2.1. Chemistry

Synthesis of the target compounds *N^4^*-(1*H*-indazol-5-yl)-*N^2^*-phenylpyrimidine-2,4-diamines **4a**–**i** was employed via the application of the chemical reactions outlined in Figure 1. Compounds **3a** and **b** were obtained via a nucleophilic substitution reaction of 5-aminoindazole **2** with 2,4-dichloropyrimidine **1a** or 5-flouro-2,4 dichloropyrimidine **1b**. The chlorine atom at position **4** of compound **1a** or **1b** was reacted with the amino group at position **5** of 5-aminoindazol **2** regio-selectively. Compounds **3a** and **b** were first synthesized by refluxing a mixture of 5-substituted-2,4-dichloropyrimidine **1a** or **1b** with 5-aminoindazole **2** using equimolar concentrations in ethanol and a few drops of HCl. However, impure compounds and poor yields were obtained using this method. Thus, another method was used by replacing HCl with triethylamine, yielding pure products with good yields [26]. The presence of the electronegative fluorine moiety of compound **1b** reduced the reaction time to around three hours, while the presence of hydrogen at the same position resulted in a longer reaction time (~12 h). Another nucleophilic substitution reaction took place to synthesize compounds **4a**–**i**, by the substitution of the Cl atom at position 2 of the pyrimidine ring in compounds **3a** and **b** with the amino group of the aniline derivatives, by refluxing the appropriate aniline with compound **3a** or **3b** in butanol and a few drops of HCl to yield the target compounds **4a**–**i**; Figure 1. All the used anilines were commercially available except the morpholino aniline, which was synthesized by reduction of the nitro group of compound **6** to afford the morpholino-aniline derivative **7**, according to the reported method [29] as presented in Figure 2. In general, target compounds having an F atom at position 5 of the pyrimidine ring were obtained faster and with a better yield, which might be due to its inductive electron-withdrawing effect. Meanwhile, poorer yields were obtained from compounds having hydrogen atoms at the same position. In addition, further mechanistic studies were performed to investigate the mode of cell death and cell cycle changes.

### 2.2. Biological Evaluation

#### 2.2.1. MTT Cytotoxicity Assay

All of the synthesized compounds were screened for their cytotoxic effect against three different human cancer cells, namely breast cancer cells (MCF-7), lung cancer cells (A549), and colorectal adenocarcinoma cells (Caco2), using the MTT cytotoxic assay [30,31]. The results are summarized in Table 1, demonstrating that five compounds—**4f**, **4i**, **4a**, **4d**, and **4g**—had potent cytotoxic activity against MCF-7 cells and were more active than the reference drug, Staurosporine. Compounds **4f**, **4i**, and **4a** demonstrated the strongest cytotoxic effect, with IC_50_ values of 1.629, 1.841, and 2.958 μM, respectively, compared to the reference (IC_50_ 8.029 μM). On the other hand, IC_50_ values of compounds **4d** and **4g** were half that of the standard, with IC_50_ values of 4.798 and 4.680 μM, respectively. The rest of the compounds showed weak cytotoxic activity against MCF-7 cells. However, only molecules **4a** and **4i** showed significant antiproliferative activity against the A549 cell line, with IC_50_ values (3.304 and 2.305 μM, respectively) two- and three-fold stronger than the standard (IC_50_ 7.35 μM). Similarly, five compounds—**4i**, **4g**, **4e**, **4d**, and **4a**—demonstrated IC_50_ values of 4.990, 6.909, 7.172, 9.632, and 10.350 μM, respectively, and exhibited potent cytotoxic activity compared to the standard (IC_50_ 11.29 μM) against Caco2 cells. Further mechanistic studies were performed toward the most promising antiproliferative candidates—compounds **4f** and **4i**—to investigate their mode of cell death and cell cycle changes. Furthermore, compounds **4f** and **4i** were chosen for further cytotoxic evaluation toward normal cells (mammary gland epithelial cell line MCF-10a) (Table 2). The results showed that their IC_50_ values (23.67 and 29.5 μM, respectively) against normal cells demonstrated marked safety profiles toward human normal cells, more than the standard drug (IC_50_ = 34.8 μM) by 10 and 5 units for **4f** and **4i**, respectively. Therefore, these compounds were more selective toward cancerous cells, with selectivity index values of IS 14.5 and 16.03, respectively, relative to the standard drug with IS 4.34.

#### 2.2.2. Cell Effects of Compounds **4f** and **4i**

An encouraging strategy for cancer therapy is targeting the cell cycle, according to the reported method [32]. In this study, compounds **4i** and **4f** were tested—using a DNA flow cytometry assay—for their effects on the cell cycle of MCF-7 cells (Figure 2). Results showed that after treating MCF-7 cells with compound **4i** or **4f** for 24 h, a slight increase in the percentage of cells in the G0-G1 phase for both compounds was observed—around 7% compared with the control (53.89%). However, compound **4i** exhibited a slight increase (3%) in the percentage of the cell population in the S phase. Simultaneously, there was a significant reduction in the percentage of the cell population in the G2/M phase—32% and 46% for cells treated with compound **4f** or **4i**, respectively, compared with controls. The decline in the proportion of cells in the G2/M phase as well as the increase in the cell population in the G0-G1 and S phases indicate a reduction in cell cycle progression in MCF-7 cells.

#### 2.2.3. Apoptosis Induction and Caspase-3/7 Activation

To assess the apoptotic potential, MCF-7 cells were treated with compound **4i** or **4f** for 24 h and then assayed for Annexin-V/PI (propidium iodide) binding according to the reported protocol [31]. Results are displayed in Figure 3, which show that, these compounds increased the percentage of Annexin V/PI-stained cells to about 47% in both early and late phases of apoptosis, compared with 1.76% of the untreated cells. At the early phase of apoptosis, the average increment for the treated groups was around 27%, while it was around 14% in late-stage apoptosis.

Caspases are essential factors in apoptotic cell death since they play a key role in maintaining homeostasis, and their activation induces apoptosis. Caspase-3/7 activity results are displayed in Figure 4. When compared with untreated controls, the levels of active caspase-3/7 expression in MCF-7 cells increased from 0.43% to 19% and 26.5% after being treated with compound **4f** or **4i**, respectively. This demonstrates the potential apoptotic effect of compounds **4f** and **4i** in MCF-7 cells.

## 3. Computational Studies

### 3.1. Molecular Dynamic and System Stability

A molecular dynamic simulation was carried out to predict the performance of the extracted compounds upon binding to the active site of protein as well as its interaction and stability through simulation [33,34]. The validation of system stability is essential to trace disrupted motions and avoid artifacts that may develop during the simulation. This study assessed Root-Mean-Square Deviation (RMSD) to measure system stability during the 20 ns simulations. The recorded average RMSD values for all frames of systems—apo-protein, **4f**-complex, and **4i**-complex systems—were 2.92 ± 0.56 Å, 2.04 ± 0.41 Å, and 2.49 ± 0.34 Å, respectively (Figure 5A). These results revealed that the 4f-bound-to-protein complex system acquired a relatively more stable conformation than the other studied systems. During MD simulation, assessing protein structural flexibility upon ligand binding is critical for examining residue behavior and its connection with the ligand [35]. Protein residue fluctuations were evaluated using the Root-Mean-Square Fluctuation (RMSF) algorithm to evaluate the effect of inhibitor binding toward the respective targets over 20 ns simulations. The computed average RMSF values were 6.24 Å, 0.99 Å, and 4.70 Å for apo-protein, **4f**-complex, and **4i**-complex systems, respectively. Overall residue fluctuations of individual systems are represented in Figure 5B. These values revealed that the **4f**-bound-to-protein complex system has a lower residue fluctuation than the other systems. ROG was determined to evaluate overall system compactness as well as stability upon ligand binding during MD simulation [36,37]. The average Rg values for apo-protein, **4f**-complex, and **4i**-complex systems were 18.25 ± 0.07 Å, 18.12 ± 0.07 Å, and 18.16 ± 0.08 Å, respectively (Figure 5C). According to the observed behavior, the 4f-complex has a highly stiff structure against caspase-3. The compactness of the protein hydrophobic core was examined by calculating the protein’s Solvent Accessible Surface Area (SASA). This was performed by measuring the surface area of the protein visible to the solvent, which is important for biomolecule stability [38]. The average SASA values for apo-protein, **4f**-complex, and **4i**-complex systems were 11,246 Å, 11,068 Å, and 11,174 Å, respectively (Figure 5D). The SASA finding, when paired with the observations from the RMSD, RMSF, and ROG computations, confirmed that the **4f**-complex system remains intact inside the S2 domain binding site of caspase-3 receptors.

### 3.2. Binding Interaction Mechanism Based on Binding Free Energy Calculation

A popular method for determining the binding free energies of small molecules to biological macromolecules is the molecular mechanics’ energy technique (MM/GBSA), which combines the generalized Born and surface area continuum solvation, and it may be more trustworthy than docking scores [39]. The MM-GBSA program in AMBER18 was used to calculate the binding free energies by extracting snapshots from the trajectories of the systems. As shown in Table 3, all the reported calculated energy components (except ΔG_solv_) gave high negative values, indicating favorable interactions. The results indicate that the binding affinities of the **4f**-complex and **4i**-complex systems were −25.56 kcal/mol and −15.63 kcal/mol, respectively.

The interactions between the **4f** and **4i** compounds and the caspase-3 receptor protein residues are driven by the more positive electrostatic energy component, as shown by a detailed examination of each energy contribution, leading to the reported binding free energies. Substantial binding free energy values were observed in the gas phase for all the inhibition processes, with values up to −88.57 and −93.65 kcal/mol, respectively (Table 3). 

### 3.3. Identification of the Critical Residues Responsible for Ligand Binding

The total energy involved when **4f** and **4i** compounds bind with these enzymes was further decomposed into the involvement of individual site residues to gain more knowledge about important residues involved in the inhibition of the S2 domain binding site receptor of caspase-3 receptors. From Figure 6, the major favorable contribution of the **4f** compound to the S2 domain binding site receptor is predominantly observed from residues Met 33 (−0.397 kcal/mol), Gly 94 (−0.598 kcal/mol), Glu 95 (−1.188 kcal/mol), Cys 135 (−1.159 kcal/mol), Arg 136 (−0.773 kcal/mol), Gly 137 (−1.183 kcal/mol), Thr 138 (−1.265 kcal/mol), Tyr 164 (−0.302 kcal/mol), Tyr 166 (−1.619 kcal/mol), Trp 167 (−0.426 kcal/mol), Arg 168 (−0.531 kcal/mol), Ser 212 (−0.459 kcal/mol), Phe 213 (−0.187kcal/mol), and Phe 217 (−0.997 kcal/mol).

On the other hand, the major favorable contribution of the **4i** compound to the S2 domain binding site receptor of caspase-3 is predominantly observed from residues Thr 34 (−0.182 kcal/mol), Glu 95 (−0.378 kcal/mol), Gly 137 (−0.625 kcal/mol), Thr 138 (−2.377 kcal/mol), Glu 139 (−1.047 kcal/mol), Gly 163 (−0.624 kcal/mol), Tyr 164 (−0.179 kcal/mol), Tyr 165 (−0.175 kcal/mol), and Phe 217 (−0.146 kcal/mol).

### 3.4. Ligand–Residue Interaction Network Profiles

One of the purposes of drug design is to make structural changes to therapeutic molecules to increase bioavailability, reduce toxicity, and improve pharmacokinetics [40].

The binding of receptor-specific active site residues to particular groups in the drug molecule results in the suppression of caspase-3, a key mediator of apoptotic cell death in mammals that cleaves over 500 cellular substrates to carry out the apoptosis program [41,42]. In light of the tight association between apoptosis and a wide range of disorders, caspase-3 inhibitors have the potential to pave the way for new treatments for immunodeficiency, Alzheimer’s, Parkinson’s, Huntington’s, ischaemia, brain trauma, and amyotrophic lateral sclerosis [43]. It has been observed that the structural interactions of both compounds are hydrophobic and electrostatic in nature in the S2 domain binding site of the caspase-3 receptor. 

Figure 7 shows that the NH group of compound **4f**’s indazole ring occupied the S2 binding pocket via a secure network of H-bonds with Gly 94 and Glu95. Furthermore, Api-pi stacking was discovered between the Tyr 165 and Phe 217 residues and the pyrimidine ring. Additionally, the hot spot Arg 186 residue produced both π-cation and π-alkyl interactions with the phenyl and morpholine rings. Ultimately, a π-cation interaction (electrostatic interaction) between Met 33 and the phenyl ring of indazole was identified (Figure 7A). Compound **4i**, on the other hand, has developed a two π-cation contact with the indazole ring. Moreover, the trimethoxy ring has formed π–π stacking with Tyr 165. Eventually, Ala 134 formed a π-alkyl interaction with the pyrazole ring of indazole (Figure 7B).

### 3.5. In Silico ADMET Properties Prediction

A compound must meet the following requirements to be considered a prospective physiologically active molecule: (1) molecular weight < 500, (2) log P (lipophilicity) < 5, (3) H-bond donors (sum of NH and OH) < 5, (4) H-bond acceptors (sum of N and O) < 10, and (5) rotatable bonds (an extra requirement proposed by Veber) < 10.

Based on the above criteria, the synthesized compounds were subjected to in silico tests for ADMET prediction for testing bioavailability and toxicity.

Table 4 shows the derived parameters for Lipinski’s rule of five, topological polar surface area, aqueous solubility, and the number of rotatable bonds. The values for human intestinal absorption ranged from 88.003231 to 92.405990%, showing that the synthesized compounds had a moderate to good absorption capacity and supported their interaction with the target cell (Table 5). 

The in vitro Caco-2 cell permeability in the range of 0.727024–48.4113 nm/s and the in vitro MDCK cell permeability in the range of 0.268974–40.1723 nm/s characterized the synthesized compounds as having high permeability. The synthesized compounds have values ranging from 83.42 to 100.00%, indicating that they have a high affinity for proteins. The in vivo blood–brain barrier penetration ranges from 0.055 to 0.86, indicating that they have a low to moderate distribution in vivo, with medium to strong penetration capacity (Table 5). Bioactivity and toxicity risk values of synthesized compounds are shown in Table 6.

## 4. Structure–Activity Relationship

Concerning the data adopted in Table 1 and Table 2, the presence of sulfadiazine of compound **4a** or the trimethoxy groups of compound **4i** at position 4 of the aniline ring resulted in its strong inhibitory activity and produced broad and potent antiproliferative activity against all of the three tested cell lines. However, the replacement of the hydrogen atom at position 5 of the pyrimidine ring in compound **4i** with fluorine in compound **4c** led to diminished activity. Meanwhile, the morpholino substituent at the *para* position of the aniline ring in compound **4f** led to significant and selective cytotoxic activity against the MCF-7 cell line. The sulfonamide substituent in compounds **4d** and **4g** caused strong cytotoxic activity against MCF-7 cells that was twofold as potent as the standard, while strong activity was observed for these two molecules against Caco2 cell lines by about double or equal to the standard, respectively. Compound **4e**, with the fluorine atom in position 5 of the pyrimidine ring along with the sulfanilamide group, illustrated potent antiproliferative activity toward the Caco2 cell line only. The sulfathiazole substituent in compounds **4h** and **4b** led to a lack of cytotoxic activity in all the three tested cell lines. 

## 5. Conclusions

The authors synthesized and identified nine new indazol-pyrimidine derivatives according to different analyses. All the new compounds were evaluated for anticancer inhibitory activity against MCF-7, A549, and Caco2 human cancer cell lines. Five compounds possessed significant cytotoxic potential against MCF-7 cells and were more potent than the reference drug. From this, compounds **4f** and **4i** exhibited the lowest IC_50_ values of 1.629 and 1.841 µM, respectively, compared with the reference drug with an IC_50_ value of 8.029 µM. In addition, five products showed cytotoxic activity stronger than the standard against Caco2 cells. Moreover, two compounds evidenced potent antiproliferative activity that was stronger than the reference against the A549 cell line. Additionally, the most active products, **4f** and **4i**, were further examined for their mechanism of action by flow cytometry assay. It turns out that they were capable of activating caspase-3/7 and, therefore, inducing apoptosis. On the other hand, these two compounds demonstrated marked safety profiles toward human normal cells (MCF-10a), more than the reference, indicating that these compounds are more selective to cancerous cells relative to the reference. Consequently, the two promising candidates will be subjected to extensive future studies for *in vivo* animal models evaluation, and they can act as new compounds in developing new potent and highly safe anticancer products. We hope to produce highly effective, low-toxicity anticancer agents after the mandatory biological studies have been performed. Following that, the interaction’s stability was assessed using a typical atomistic 20 ns dynamic simulation study. A number of parameters derived from MD simulation trajectories were computed and validated for the protein–ligand complex’s stability under dynamic conditions. Prediction of computational drug-like properties showed that most of the synthesized compounds are safe with acceptable ADMET and druggable properties.

## 6. Experimental Section

### 6.1. Chemistry

All reagents and solvents were obtained from commercial suppliers and were used without further purification. When necessary, solvents were dried by standard methods. Melting points (°C) were measured in open-glass capillaries using Branstead 9100 electrothermal melting point apparatus and are uncorrected. NMR spectra were obtained on a Bruker AC 500 Ultra Shield NMR spectrometer (Fällanden, Switzerland) at 500.13 MHz or (700) for ^1^H and 125.76 MHz for ^13^C; the chemical shifts are expressed in δ (ppm) downfield from tetramethylsilane (TMS) as internal standard at 154 MHz, and coupling constants *(J*) are expressed in Hz. Deuteriodimethylsulphoxide (DMSO-*d*_6_) was used as a solvent. The splitting patterns were designated as s (singlet), d (doublet), t (triplet), m (multiplet), and br. s (broad singlet). Electrospray Ionization Mass Spectra (ESI-MS) were recorded on an Agilent 6410 Triple Quad Tandem Mass Spectrometer at 4.0 and 3.5 kV for positive and negative ions, respectively. High-Resolution Mass Spectra (HR-MS) were recorded on JEOL JMS-700 using Electron Impact (EI) ionization mode by keeping ionization energy at 70 eV. Elemental analyses (C, H, and N) were conducted at the Micro Analytical Center of the Faculty of Science of Cairo University, Cairo, Egypt. They aligned with the proposed structures within ±0.1–0.2%. 

#### 6.1.1. General Method for the Synthesis of *N*-(2-Chloro-5-substituted pyrimidin-4-yl)-1*H* -indazol-5-amine (Compounds **3a** and **b**)

A mixture of 2,4 dichloropyrimidine or 5-flouro-2,4 dichloropyrimidine (0.027 mol) and 5-aminoindazole (3.59 g, 0.027 mol) was dissolved in (8 mL) Ethanol with continuous stirring. Triethylamine (2.7 g, 0.027 mol) was added gradually, followed by refluxing the mixture at 80 °C for 4–6 h. After completion of the reaction (which was monitored by TLC), the formed precipitate was filtered off, washed with cold water, dried, and recrystallized from ethanol to afford 5-substituted-N-(2-chloropyrimidin-4-yl)-1H-indazol-5-amine according to the reported method [26].

#### 6.1.2. General Procedure for Preparation of Compounds **4a**–**i**

To a mixture of compound **3** (0.0018 mol) in butanol (25 mL), the appropriate aniline derivative (0.0018 mol) was added, followed by the addition of 4 drops of conc. HCl. The mixture was refluxed overnight, and after cooling, the formed precipitate was filtered off, washed with hot ethanol and/or ethyl acetate, and filtered off while hot, then recrystallized from ethanol to afford the desired compounds **4a**–**i**.

*4-((4-((1H-indazol-5-yl)amino)-5-fluoropyrimidin-2-yl)amino)-N-(pyrimidin-2-yl)benzenesulfonamide* **4a**, Yield: 68%; m.p.: 343–345 °C; IR (υ_max_/cm^−1^): 3373–3323 (4NH), 3143 (CH, aromatic), 1350, 1155 (SO_2_); ^1^H-NMR (DMSO-*d*_6_ *δ* ppm): 5.56 (br, s, 2H, 2NH), 7.01 (br, 1H, pyrimidine, CH-5), 7.55 (d, *J* = 14 Hz, 1H, indazole, CH-6), 7.62 (d, *J* = 7 Hz, 1H, indazole, CH-7), 7.68 (d, *J* = 14 Hz, 2H, ph, CH-3,5), 7.77 (d, *J* = 7 Hz, 2H, ph, CH-2, 6), 8.07 (s b, 1H, indazole, CH-4), 8.09 (s, 1H, indazole, CH-3), 8.3 (s br, 1H, F-pyrimidine), 8.4 (br, 2H, pyrimidine, CH-4,6), 10.6, 10.7 (2s, 2H, 2NH); ^13^C-NMR (DMSO-*d*_6_, *δ* ppm): 110.8, 113.4, 115.4, 116.2, 119.2, 123.1, 124.0, 130.0, 130.2, 133.9, 134.1, 138.5, 139.5, 141.0, 142.9, 151.6, 152.67, 152.74, 157.37, 157.6, 158.8 (Ar-C); MS, *m*/*z* (%): 476 (M − 1) (20), 475 (M − 2) (6), consistent with the molecular formula C_21_H_16_FN_9_O_2_S. 

*4-((4-((1H-indazol-5-yl)amino)-5-fluoropyrimidin-2-yl)amino)-N-(thiazol-2-yl)benzenesulfonamide* **4b**, Yield: 72%; m.p.: 285–287 °C; IR (υ_max_/cm^−1^): 3320–3209 (4NH), 3057 (CH, aromatic), 1327, 1141 (SO_2_); ^1^H NMR (DMSO-*d*_6_, *δ* ppm): 5.8 (br, s, 2H, 2NH), 6.79–7.61 (m, 8H, Ar), 8.01 (s, 1H, indazole, CH-3), 8.09 (br, 1H, indazol, CH-7), 8.3 (br, 1H, F-pyrimidine), 10.8, 10.9 (2s, 2H, 2NH); ^13^C-NMR (DMSO-*d*_6_, *δ* ppm): 108.3, 110.4, 115.6, 119.8, 122.7, 123.8, 124.6, 126.8, 127.7, 129.3, 133.6, 136.6, 138.2, 138.8, 140.3, 141.1, 150.5, 152.7, 152.8, 168.8 (Ar-C); MS, *m*/*z* (%): 483 (M + 1) (6), consistent with the molecular formula C_20_H_15_FN_8_O_2_S_2_. 

*5-fluoro-N4-(1H-indazol-5-yl)-N2-(3,4,5-trimethoxyphenyl)pyrimidine-2,4-diamine* **4c**, Yield: 75%; m.p.: 352–354 °C; IR (υ_max_/cm^−1^): 3350–3317 (3NH), 3095 (CH, aromatic); ^1^H NMR (DMSO-*d*_6_, *δ* ppm): 3.5 (s, 3H, OCH_3_ at C-4), 3.6 (s, 6H, 2OCH_3_ at C-3,5), 6.87 (s, 2H, ph, CH-2,6), 7.50 (d, *J* = 8.9 Hz, 1H, indazol, CH-6), 7.56 (s, 1H, indazole, CH-4), 7.9–8.18 (m, 3H, Ar), 9.6, 10.03, 13.1 (3s br, 3H, 3NH); ^13^C-NMR (DMSO-*d*_6_, *δ* ppm): 56.0, 56.5 (2OCH_3_, ph, C-3,5), 60.5 (OCH_3_, ph, C-4) 99.25 (2C, ph, CH-2,6), 110.6, 112.9, 113.8, 123.0, 123.1,123.3, 123.3, 131.1, 133.8, 138.0, 153.2, 153.8, 170.84 (Ar-C); MS, *m*/*z* (%): 412.5 (M + 2) (65), 413.3 (M + 3) (100), consistent with the molecular formula C_20_H_19_FN_6_O_3_.

*3-((4-((1H-indazol-5-yl)amino)-5-fluoropyrimidin-2-yl)amino)benzenesulfonamide* **4d**, Yield: 63%; m.p.: 286–288 °C; IR (υ_max_/cm^−1^): 3449 (NH_2_), 3373–3250 (3NH), 3143 (CH, aromatic), 1350, 1155 (SO_2_); ^1^H NMR (DMSO-*d*_6_, *δ* ppm): 7.2 (s, 2H, NH_2_), 7.3 (t, *J* = 6 Hz, 1H, ph, CH-5), 7.36 (d, *J* = 7 Hz, 1H, ph, CH-4), 7.53 (d, *J* = 7 Hz, 1H, indazole, CH-6), 7.6 (s, 1H, indazole, CH-4), 8.00 (d, *J* = 7 Hz, 1H, ph, CH-6), 8.03 (s, 1H, ph, CH-2), 8.09 (s, 1H, indazole, CH-3), 8.012 (d, *J* = 7 Hz, 1H, indazol, CH-7), 8.2 (s, 1H, F-pyrimidine), 9.4, 9.5, 13.02 (3s, 3H, 3NH); 13C-NMR (DMSO-*d*_6_, *δ* ppm): 110.2, 113.3, 113.9, 117.3, 120.0, 122.6, 123.0, 123.4, 129.2, 130.0, 133.4, 137.7, 139.0, 139.3, 140.7, 144.5, 151.6 (Ar-C); MS, *m*/*z* (%): 397 (M − 2) (20), consistent with the molecular formula C_17_H_14_FN_7_O_2_S.

*4-((4-((1H-indazol-5-yl)amino)-5-fluoropyrimidin-2-yl)amino)benzenesulfonamide* **4e**, Yield: 65%; m.p.: 303–305 °C; IR (υ_max_/cm^−1^): 3433 (NH_2_), 3352–3230 (3NH), 3133 (CH, aromatic), 1328, 1155 (SO_2_); ^1^H NMR (DMSO-*d*_6_, *δ* ppm): 7.13 (s, 2 H, NH_2_), 7.56–7.61 (m, 4H, Ar-H), 7.81 (s, 1H, indazole, CH-4), 7.83 (d, *J* = 4.9 Hz, 1H, indazol, CH-6), 8.06 (s, 1H, indazole, CH-3), 8.15 (d, *J* = 3.6 Hz, 1H, Indazol, CH-7), 8.16 (s br, 1H, F-pyrimidine), 9.50, 9.62, 13.08 (3s, 3H, 3NH); ^13^C-NMR (DMSO-*d*_6_, *δ* ppm): 110.3, 113.4, 117.6, 123.2, 123.5, 126.7, 131.7, 133.6, 135.7, 137.6, 140.3, 140.4, 140.6, 142.0, 144.4, 150.7, 155.3 (Ar-C); MS, *m*/*z* (%): 398 (M − 1) (10), 397 (M − 2) (14), consistent with the molecular formula C_17_H_14_FN_7_O_2_S. 

*5-fluoro-N4-(1H-indazol-5-yl)-N2-(4-morpholinophenyl) pyrimidine-2,4-diamine* **4f**, Yield: 74%; m.p.: 298–300 °C; IR (υ_max_/cm^−1^): 3315–3290 (3NH), 3012 (CH, aromatic); ^1^H NMR (DMSO-*d*_6_, *δ* ppm): 3.1 (m, 4H, 2CH_2_, morpholine, C-2,6), 3.7 (m, 4H, 2CH_2_, morpholine, C-3,5), 7.4 (d, *J* = 7 Hz, 1H, indazol, CH-6), 7.5 (m, 3H, Ar), 8.0–8.27 (m, 4H, Ar), 8.3 (s, 1H, F-pyrimidine), 10.0, 10.5, 10.8 (3s, 3H, 3NH); ^13^C-NMR (DMSO-*d*_6_, *δ* ppm): 56.5 (2CH_2_, morpholine, C-2,6), 65.5 (2CH_2_, morpholine, C-3,5), 110.8, 113.8, 115.4, 122.9, 123.1, 123.4, 123.7, 129.6, 130.8, 133.9, 134.0, 138.1, 138.5, 141.4, 145.0, 146.4, 152.1, 153.1, 153.5, (Ar-C); MS, *m*/*z* (%): 406 (M + 1) (20), consistent with the molecular formula C_21_H_20_FN_7_O.

*4-((4-((1H-indazol-5-yl) amino) pyrimidin-2-yl) amino) benzenesulfonamide.* **4g**, Yield: 66%; m.p.; 228–230 °C; IR (υ_max_/cm^−1^): 3483 (NH_2_), 3365–3315 (3NH), 3140 (CH, aromatic), 1321, 1184 (SO_2_); ^1^H NMR (DMSO-*d*_6_, *δ* ppm): 6.4 (s, 1H, NH), 7.3 (s, 2H, NH_2_), 7.46 (d, *J* = 7 Hz, 1H, pyrimidine, CH-5), 7.6 (d, *J* = 7 Hz, 1H, indazol, CH-6), 7.7 (m, 5H, Ar), 8.02–8.11 (m, 3H, Ar), 10.7, 13.1 (2s, 2H, 2NH); ^13^C-NMR (DMSO-*d*_6_, *δ* ppm): 96.6, 101.2, 114.3, 116.5, 116.62, 116.67, 124.8, 127.9, 128.2, 128.5, 129.7, 132.6, 141.1, 146.3, 158.6, 168.4, 170.0, (Ar-C); MS, *m*/*z* (%): 381.02 (M^+^) (10), 382.05 (M + 1) (55), 383 (M + 2) (20), consistent with the molecular formula C_17_H_15_N_7_O_2_S.

*4-((4-((1H-indazol-5-yl)amino)pyrimidin-2-yl)amino)-N-(thiazol-2-yl)benzenesulfonamide* **4h**, Yield: 59%; m.p.: 211–213 °C; IR (υ_max_/cm^−1^): 3290–2372 (4NH), 3143 (CH, aromatic), 1328, 1138 (SO_2_); ^1^H NMR (DMSO-*d*_6,_ *δ* ppm): 6.6–6.8 (m, 2H, Ar), 7.2 (s, 1H, indazole, CH-4), 7.53–7.7 (m, 6H, Ar), 8.05–8.08 (m, 3H, Ar-H), 9.4, 11.2, 11.4, 12.8 (4s, 4H, 4NH); ^13^C-NMR (DMSO-*d*_6_, δ ppm): 100.7, 108.5, 110.9, 114.3, 121.8, 123.0, 125.0, 127.2, 127.4, 128.1, 130.3, 134.0, 138.3, 140.6, 142.8, 152.2, 161.5, 169.2, 169.3 (Ar-C); MS, *m*/*z* (%): 463 (M − 1) (15), 466 (M + 2) (4), consistent with the molecular formula C_20_H_16_N_8_O_2_S_2._

*N4-(1H-indazol-5-yl)-N2-(3,4,5-trimethoxyphenyl) pyrimidine-2,4-diamine* **4i**, Yield: 57%; m.p.: 202–204 °C; IR (υ_max_/cm^−1^): 3340–3322 (3NH), 3101 (CH, aromatic); ^1^H NMR (DMSO-*d*_6,_ *δ* ppm): 3.63 (s, 3H, OCH_3_ at ph-C-4), 3.69 (s, 6H, 2OCH_3_, ph-C-3,5), 6.5 (s, 1H, indazol, CH-4), 6.8 (s, 2H, ph, CH-2,6), 7.48–8.2 (m, 5H, Ar), 10.3, 10.9, 13.1 (3s, 3H, 3NH); ^13^C-NMR (DMSO-*d*_6_, δ ppm): 56.0, 56.5 (2OCH_3_, ph, C-3,5), 60.5 (OCH_3_, ph, C-4), 99.2 (2C, ph, CH-2,6), 101.2, 110.9, 113.8, 123.0, 123.1,123.3, 123.3, 131.1, 133.8, 138.0, 153.2, 153.8, 170.84 (Ar-C); MS, *m*/*z* (%): 392 (M^+^) (20), 393 (M + 1) (96), consistent with the molecular formula C_20_H_20_N_6_O_3_.

### 6.2. Biological Assays

#### 6.2.1. MTT Cytotoxicity Assay

The synthesized compounds **4a**–**i** were added to the tested cells MCF-7, A549, and Caco2 using concentrations ranging from 0.1 to 10 µM for 48 h. Then, 3-[4,5-dimethylthiazol-2-yl]-2,5-diphenyl tetrazolium bromide (MTT) was added. The plates were incubated for 3 h before being read with Wallac Victor2 1420 multilabel counter in fluorescence mode at the wavelength (460/590 nm). Molecule concentrations needed to inhibit 50% of cell growth (IC_50_) were calculated, and Staurosporine was used as a positive control.

#### 6.2.2. Caspase-3/7 Assay 

Caspase-3/7 activity was calculated using Caspase-3/7 Green Flow Cytometry Assay Kit Catalog # C10427.

#### 6.2.3. Cell Cycle Analysis

Apoptosis was measured using an Annexin V-FITC Apoptosis Detection Kit and analyzed using FACSCalibur flow cytometer.

### 6.3. Molecular Dynamic Study

#### 6.3.1. System Preparation and Molecular Docking

The crystal structure of human caspase-3 was assessed at a resolution of 2.80 Å, which was retrieved from the protein data bank with codes 1GFW [44] and prepared using UCSF Chimera [45]. Using PROPKA, pH was fixed and optimized to 7.5 [46]. The extracted 2D structure was drawn using ChemBioDraw Ultra 12.1 [47]. The steepest descent approach and MMFF94 force field in Avogadro software [48] were used to optimize the 2D structure for energy minimization. In preparation for docking, hydrogen atoms were removed using UCSF chimera [45].

#### 6.3.2. Molecular Docking

AutoDock Vina was used for docking calculations [49], and Gasteiger partial charges [50] were allocated during docking. The AutoDock graphical user interface offered by MGL tools was used to outline the AutoDock atom types [51]. The grid box was determined with grid parameters x = −36.6310, y = 37.0493, and z = 31.466 for the dimension and x = 15.9631, y = 14.398, and z = 10 for the central grid and exhaustiveness = 8. The Lamarckian genetic algorithm [52] was used to create docked conformations in descending order based on their docking energy.

#### 6.3.3. Molecular Dynamic (MD) Simulations

The integration of Molecular Dynamic (MD) simulations in biological system studies enables exploring the physical motion of atoms and molecules that cannot be easily accessed by any other means [53]. The insight extracted from performing this simulation provides an intricate perspective into the biological systems’ dynamic evolution, such as conformational changes and molecule association [53]. The MD simulations of all systems were performed using the GPU version of the PMEMD engine present in the AMBER 18 package [54]. The partial atomic charge of each compound was calculated with ANTECHAMBER’s General Amber Force Field (GAFF) technique [55]. The Leap module of the AMBER 18 package implicitly solvated each system within an orthorhombic box of TIP3P water molecules within 10 Å of any box edge. The leap module was used to neutralize each system by incorporating Na^+^ and Cl^−^ counterions. A 2000-step initial minimization of each system was carried out in the presence of a 500 kcal/mol applied restraint potential, followed by a 1000-step full minimization using the conjugate gradient algorithm without restraints. During the MD simulation, each system was gradually heated from 0 K to 300 K over 500 ps, ensuring that all systems had the same amount of atoms and volume. The system’s solutes were subjected to a 10 kcal/mol potential harmonic constraint and a 1 ps collision frequency. Following that, each system was heated and equilibrated for 500 ps at a constant temperature of 300 K. To simulate an isobaric-isothermal (NPT) ensemble, the number of atoms and pressure within each system for each production simulation were kept constant, with the system’s pressure maintained at 1 bar using the Berendsen barostat [56]. For 20 ns, each system was MD simulated. The SHAKE method was used to constrain the hydrogen bond atoms in each simulation. Each simulation used a 2fs step size and integrated an SPFP precision model. An isobaric-isothermal ensemble (NPT) with randomized seeding, constant pressure of 1 bar, pressure-coupling constant of 2 ps, temperature of 300 K, and a Langevin thermostat with a collision frequency of 1 ps was used in the simulations.

#### 6.3.4. Post-MD Analysis

After saving the trajectories obtained by MD simulations every 1 ps, the trajectories were analyzed using the AMBER18 suite’s CPPTRAJ [57] module. The Origin [58] data analysis program and Chimera [45] were used to create all graphs and visualizations.

#### 6.3.5. Thermodynamic Calculation

The Poisson-Boltzmann or generalized Born and surface area continuum solvation (MM/PBSA and MM/GBSA) approach is useful in the estimation of ligand-binding affinities [59,60,61]. The Protein–Ligand complex molecular simulations used by MM/GBSA and MM/PBSA compute rigorous statistical-mechanical binding free energy within a defined force field.

Binding free energy averaged over 200 snapshots extracted from the entire 20 ns trajectory. The estimation of the change in binding free energy (ΔG) for each molecular species (complex, ligand, and receptor) can be represented as follows: [62].
(1)$$\Delta G_{\mathrm{bind}}=G_{\mathrm{complex}}-G_{\mathrm{receptor}}-G_{\mathrm{ligand}}$$
(2)$$\Delta G_{\mathrm{bind}}=E_{\mathrm{gas}}+G_{\mathrm{sol}}-\mathrm{TS}$$
(3)$$E_{\mathrm{gas}}=E_{\mathrm{int}}+E_{\mathrm{vdw}}+E_{\mathrm{ele}}\;$$
(4)$$G_{\mathrm{sol}}=G_{\mathrm{GB}}+G_{\mathrm{SA}}\;$$
(5)$$G_{\mathrm{SA}}=\gamma \mathrm{SASA}$$

The terms E_gas_, E_int_, E_ele_, and E_vdw_ symbolize the gas–phase energy, internal energy, Coulomb energy, and van der Waals energy. The E_gas_ was directly assessed from the FF14SB force field terms. Solvation free energy (G_sol_) was evaluated from the energy involvement from the polar states (G_GB_) and non-polar states (G). The non-polar solvation free energy (G_SA_) was determined from the Solvent Accessible Surface Area (SASA) [63,64] using a water probe radius of 1.4 Å. In contrast, solving the GB equation assessed the polar solvation (G_GB_) contribution. Items S and T symbolize the total entropy of the solute and temperature, respectively. The MM/GBSA-binding free energy method in Amber18 was used to calculate the contribution of each residue to the total binding free energy.

#### 6.3.6. Computation of Drug-like Parameters and ADMET Profiling

The online tool kit Molinspiration (http://www.molinspiration.com/ (accessed on 16 April 2023)) and the OSIRIS property explorer were used to compute drug-like features from the above-mentioned compounds’ 2D chemical structures [65,66,67].

Pre-ADMET online server (https://preadmet.bmdrc.kr/ (accessed on 16 April 2023)) was used for calculating pharmacokinetic parameters such as adsorption, distribution, metabolism, excretion, and some of the computed properties such as human intestinal absorption (HIA%), Caco2 cell permeability (nm/s), MDCK (Medin-Darbey Canine Kidney Epithelial Cells) cell permeability (nm/s), plasma protein binding (%), blood–brain barrier penetration (C. brain/C. blood), and Pgp inhibition [68]. Molinspiration’s (http://www.molinspiration.com/ (accessed on 16 April 2023)) online tool kit predicted the bioactivity of synthesized compounds, and OSIRIS property explorer estimated toxicity characteristics such as mutagenicity, tumorigenicity, irritating effects, and reproductive impacts [69].

## Data Availability

Not applicable.

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
