# Peer review of "New Indazol-Pyrimidine-Based Derivatives as Selective Anticancer Agents: Design, Synthesis, and In Silico Studies"

_molecules, 2023, doi:10.3390/molecules28093664_

Round 1

Reviewer 1 Report

The references (28,39) cited in the chemistry part are not fitting to the reactions described.Please correct.
In scheme 1, 2a should be 1b.
In the experimental part 6.1.1, compound 3a has already been described. Are the properties similar to the described compound (give the correct reference)? If 3b is new then it has to be described. If not the right reference should be given.

Reviewer 2 Report

The presented work concerns the synthesis, biological research and in silico studies of a series of indazol-pyrimidine derivatives. Chemically, the synthesis is simple and basically one-step. The obtained compounds were characterized by spectroscopic methods and by means of mass spectrometry. It would be helpful to unambiguously determine the structure of at least one of the series of compounds using X-ray crystallography. This part of the work can only be assessed as moderately interesting. Studies on biological activity have yielded good results for several derivatives. Importantly, the toxicity to healthy cells was also determined.

It is not clear whether the test compounds had medically useful solubility in physiological medium or only in DMSO. Information on this (line 456 onwards is insufficient)

In several places the graphics are of unacceptable quality, e.g. formula in Table 1 (line 137), Table 2 (line 140).

All in all, the presented material can be considered for publication, although it is not a breakthrough in the medicinal chemistry of anti-cancer substances.

Author Response

  • The presented work concerns the synthesis, biological research and in silico studies of a series of indazol-pyrimidine derivatives. Chemically, the synthesis is simple and basically one-step. The obtained compounds were characterized by spectroscopic methods and by means of mass spectrometry. It would be helpful to unambiguously determine the structure of at least one of the series of compounds using X-ray crystallography. This part of the work can only be assessed as moderately interesting. Studies on biological activity have yielded good results for several derivatives. Importantly, the toxicity to healthy cells was also determined.
  • Unfortunately, the X-ray apparatus was not working, till now. In a future work and after working up the apparatus, we will determine the structure using X-ray
  • It is not clear whether the test compounds had medically useful solubility in physiological medium or only in DMSO. Information on this (line 456 onwards is insufficient)
  • Our study was in vitro So the solubility of the compounds was detected in DMSO only. Further study will be carried out in future to study the effect of the most potent compounds in vivo
  • In several places the graphics are of unacceptable quality, e.g. formula in Table 1 (line 137), Table 2 (line 140).
  • The required corrections have been done
  • All in all, the presented material can be considered for publication, although it is not a breakthrough in the medicinal chemistry of anti-cancer substances.

Reviewer 3 Report

In this manuscript the authors have been synthesized potent new anticancer agents indazol-pyrimidine derivatives based. All the prepared compounds, tested for in vitro cell line showed inhibitory activities against three different cancerous cells. Results demonstrated that, five of the some of novel compounds possessed significant cytotoxic inhibitory activity against MCF demonstrating promising suppression power. Experimental section is well written and all the data reported are consistent. Thus I recomend publication without any revision.

Reviewer 4 Report

In this manuscript entitled: New Indazol-Pyrimidine-based derivatives as selective anticancer agents: design, synthesis, in silico studies, the authors: Hanaa M. Al-Tuwaijri *, Ebtehal S. Al-Abdullah, Ahmed A. El-Rashedy, Siddique Akber Ansari, Aliyah Al-momen, Hanan M. Alshibl, Mogedda E. Haiba and Hamad M. Alkahtani present the synthesis of some indazol-pyrimidine derivatives and evaluation of their anticancer activity.

I believe that the paper may be suitable for publication in the MDPI journal Molecules after addressing the following considerations listed below.

- I suggest including in Introduction section other recent references.

- In line 17, please write the IC50 values with three decimal places.

- In lines 18 and 19, is need that “4i, 4g, 4e, 4d and 4e” be corrected to “4i, 4g, 4e, 4d, and 4a”.

- In lines 82 and 83, is need that N4-(1H-indazol-5-yl)-N2-phenylpyrimidine-2,4-diamine” be corrected to N4-(1H-indazol-5-yl)-N2-phenylpyrimidine-2,4-diamines”.

- In line 91, is necessary that “HCL” be corrected to “HCl.

- In Scheme 1, is need that 2a, R1 = F” be corrected to “1b, R1 = F”.

- In Scheme 1, R1 and R2 is need to be specified for the new derivatives 4ai.

- I suggest that the synthesis of intermediate 7 be presented in another scheme.

- All headings should be written using the title case, e.g. in line 109, I suggest writing ”2.2. Biological Evaluation” instead of “2.2. Biological evaluation”.

- In Table 1, I suggest that the general structure of compounds 4a-i to be removed or if it is maintained, it is need to be corrected and mentioned under it “4ai".

- In Table 1, for R2 I suggest writing:

N-(pyrimidin-2-yl)sulfonamido” instead of “N-(pyrimidin-2-yl) sulfonamide”,

N-(thiazol-2-yl)sulfonamido”instead of “N-(thiazol-2-yl) sulfonamide”, and

4-morpholino” instead of “Morpholine”.

- In Table 1, please replace “IC50 uM” with “IC50 in μM”.

- In line 352, I need that “Deuteriodimethylsulphoxide (DMSO-d6) were used as solvents.” be corrected to “Deuteriodimethylsulphoxide (DMSO-d6) was used as solvent.”

- In the names of compounds 4a (lines 376 and 377) and 4b (lines 387 and 388), the space between "benzene" and "sulfonamide" is not necessary.

- In line 395, I suggest that “5-fluoro-N4-(1H-indazol-5-yl)-N2-(3,4,5-trimethoxyphenyl)pyrimidine-2,4-diamineto be written “5-fluoro-N4-(1H-indazol-5-yl)-N2-(3,4,5-trimethoxyphenyl)pyrimidine-2,4-diamine”.

- In line 421, I suggest writing “5-fluoro-N4-(1H-indazol-5-yl)-N2-(4-morpholinophenyl)pyrimidine-2,4-diamine” instead of “5-fluoro-N4-(1H-indazol-5-yl)-N2-(4-morpholinophenyl) pyrimidine-2,4-diamine”.

- In line 429, I need that “C21H20FN7O” be corrected to “C21H20FN7O”.

- In line 430, I suggest writing “4-((4-((1H-indazol-5-yl)amino)pyrimidin-2-yl)amino)benzenesulfonamide” instead of “4-((4-((1H-indazol-5-yl) amino) pyrimidin-2-yl) amino) benzenesulfonamide”.

- In line 446, I suggest writing “N4-(1H-indazol-5-yl)-N2-(3,4,5-trimethoxyphenyl)pyrimidine-2,4-diamine” instead of “N4-(1H-indazol-5-yl)-N2-(3,4,5-trimethoxyphenyl) pyrimidine-2,4-diamine”.

- Please verify in line 452, the m/z “ 392 (M+2)”. The Mr of compound 4i is 392.

- I think that the order of references 16 and 17 should be reversed.

- In line 616, for reference 31, please add the title of the article: “Design, Synthesis, Anticancer Evaluation, Enzymatic Assays, and a Molecular Modeling Study of Novel Pyrazole–Indole Hybrids“.

- In the supplementary material, please include the IR, NMR and MS spectra.

Round 2

Reviewer 4 Report

I believe that the manuscript entitled manuscript entitled: New Indazol-Pyrimidine-Based Derivatives as Selective Anticancer Agents: Design, Synthesis, In Silico Studies by Hanaa M. Al-Tuwaijri *, Ebtehal S. Al-Abdullah, Ahmed A. El-Rashedy, Siddique Akber Ansari, Aliyah Al-momen, Hanan M. Alshibl, Mogedda E. Haiba and Hamad M. Alkahtani has been improved and is suitable for publication in the MDPI journal Molecules.

In Scheme 1, for R2 please write:

N-(pyrimidin-2-yl)sulfonamido” instead of “N-(pyrimidin-2-yl) sulfonamide”,

N-(thiazol-2-yl)sulfonamido”instead of “N-(thiazol-2-yl) sulfonamide”, and

4-morpholino” instead of “morpholine”.

- In line 398, I suggest that “5-fluoro-N4-(1H-indazol-5-yl)-N2-(3,4,5-trimethoxyphenyl)pyrimidine-2,4-diamine 4cto be written 5-fluoro-N4-(1H-indazol-5-yl)-N2-(3,4,5-trimethoxyphenyl)pyrimidine-2,4-diamine 4c”.

- In line 424, I suggest writing “5-fluoro-N4-(1H-indazol-5-yl)-N2-(4-morpholinophenyl)pyrimidine-2,4-diamine 4f instead of “5-fluoro-N4-(1H-indazol-5-yl)-N2-(4-morpholinophenyl) pyrimidine-2,4-diamine 4f”.

- In line 433, I suggest writing “4-((4-((1H-indazol-5-yl)amino)pyrimidin-2-yl)amino)benzenesulfonamide 4ginstead of “4-((4-((1H-indazol-5-yl) amino) pyrimidin-2-yl) amino) benzenesulfonamide. 4g”.

- In line 449, I suggest writing “N4-(1H-indazol-5-yl)-N2-(3,4,5-trimethoxyphenyl)pyrimidine-2,4-diamine 4iinstead of “N4-(1H-indazol-5-yl)-N2-(3,4,5-trimethoxyphenyl) pyrimidine-2,4-diamine 4i”.

- In line 623, for reference 31, please add the title of the article: “Design, Synthesis, Anticancer Evaluation, Enzymatic Assays, and a Molecular Modeling Study of Novel Pyrazole–Indole Hybrids“.